# The Difficulties in Emotional Regulation among a Cohort of Females with Lipedema

**DOI:** 10.3390/ijerph192013679

**Published:** 2022-10-21

**Authors:** Mohammad Al-Wardat, Chantelle Clarke, Nuha Alwardat, Manal Kassab, Chiara Salimei, Paola Gualtieri, Marco Marchetti, Talitha Best, Laura Di Renzo

**Affiliations:** 1Department of Rehabilitation Sciences, Faculty of Applied Medical Sciences, Jordan University of Science and Technology, Irbid P.O. Box 3030, Jordan; 2NeuroHealth Lab, Appleton Institute, School of Health, Medical and Applied Sciences, CQUniversity Australia, Brisbane 4000, Australia; 3Faculty of Applied Medical Sciences, Jerash University, Jerash P.O. Box 311, Jordan; 4Department of Maternal and Child Health, Faculty of Nursing, Jordan University of Science and Technology, Irbid P.O. Box 3030, Jordan; 5Department of Clinical Sciences and Translational Medicine, University of Tor Vergata, Via Montpellier 1, 00133 Rome, Italy; 6Section of Clinical Nutrition and Nutrigenomic, Department of Biomedicine and Prevention, University of Tor Vergata, Via Montpellier 1, 00133 Rome, Italy; 7PhD School of Applied Medical-Surgical Sciences, University of Rome Tor Vergata, Via Montpellier 1, 00133 Rome, Italy

**Keywords:** lipedema, emotion regulation, psychological disorders, anxiety, obesity

## Abstract

Background: Lipedema is a chronic and progressive adipose tissue disorder that causes significant morbidity and negatively influences mental health and quality of life, and increases the risk of depression, anxiety, and eating disorders. One construct of relevance to better understanding psychological disorders is emotion regulation (ER). Therefore, the aim of this study is to investigate the difficulties in ER among lipedema patients compared to healthy people without lipedema. Methods: This cross-sectional study assessed differences in ER and anxiety between two groups: 26 female patients with lipedema and 26 sex- and age-matched healthy controls. The Difficulties in Emotion Regulation Scale (DERS) assessed emotional regulation across six dimensions: Impulse control, goal-directed behavior, awareness, clarity, non-acceptance, and strategies. Anxiety was assessed by the Hamilton Anxiety Scale (HAM-A). ANOVA assessed differences in measures between lipedema and healthy control groups. Results: Lipedema patients presented with significantly more difficulties in ER and a higher level of anxiety than those without lipedema. Specifically, the lipedema group showed higher and significant differences in total DERS and anxiety scores and all DERS subscales scores compared to those without lipedema. Conclusions: Lipedema patients showed significant difficulties with ER, and were associated with anxiety symptoms, indicating that ER difficulties may play a role in developing emotional disorders, such as anxiety, for patients with lipedema. The health care provider should pay more attention to ER difficulties and psychological status among lipedema patients.

## 1. Introduction

Lipedema is a chronic, progressive, and disabling adipose tissue disorder that primarily affects women [1]. The common characteristics of lipedema are symmetric bilateral swelling, enlargement of the legs, and painful subcutaneous adipose tissue [2]. Whilst the pathological causes of lipedema are little understood, lipedema often presents in women during puberty and other times of hormonal change, such as childbirth or menopause [1]. Despite these symptoms, lipedema is still underdiagnosed or misdiagnosed as obesity or lymphedema, with significant delays in diagnosis [1]. For example, a recent study by Fetzer and Warrilow (2021) reported a median of 26–40 years to diagnosis in the UK. There is an extremely high demand for accurate diagnosis and treatment since lipedema causes significant morbidity and negatively influences mental health and quality of life (QoL) [1,3]. 

Patients with lipedema risk developing psychological disorders, such as anxiety and depression [3,4,5]. For example, Erbacher and Bertsch (2020) found that of 150 lipedema patients in a European clinic, 36.7% experienced at least one psychological disorder (anxiety, depression, eating disorders, post-traumatic stress disorder, and panic disorders). This is perhaps unsurprising when considering the difficulties those with lipedema face in gaining access to adequate health care (in both diagnosis and treatment) and unmanageable changes in body appearance over time that lead to not only reduced physical and work capability but exposure to weight stigma and reduced social functioning and isolation that can lead to these emotional or psychological disorders [1,6,7,8,9,10]. Research addressing mental health in lipedema is vital, with research demonstrating that mental health is considered as important to those with lipedema as their physical health and can be more impaired than physical health on measures of life [11,12]. Thus, lipedema’s psychological, social, and medical consequences may increase the risk of depression, anxiety, and eating disorders [1,3,7]. One construct of relevance to better understanding psychological disorders is emotion regulation (ER). 

ER controls emotional experience and expression and the ability to experience and differentiate a full range of emotions [13,14,15]. Adaptive ER modulates the intensity of emotions (as opposed to eliminating negative emotions) through characteristics such as awareness and understanding of emotions, acceptance, inhibition of impulsive behaviors and utilizing appropriate self-regulation strategies and goal-directed behaviors when experiencing negative emotions to achieve desired goals. Without these characteristics, it becomes difficult to regulate the intensity of negative emotions that may increase. Previous studies have demonstrated a direct relationship between emotional dysregulations and different types of psychopathology, such as substance abuse and eating disorders, depression, and particularly anxiety [16,17,18,19]. For example, research has shown that in health conditions associated with visual differences, such as that acne and severe obesity, emotional regulation has been shown to have moderate relationships to depression and moderate to strong relationships with anxiety [20,21]. However, the current literature lacks a specific and detailed assessment of ER difficulties and anxiety in patients with lipedema. 

Since difficulties in ER can contribute to developing psychological disorders and lipedema patients are highly vulnerable to developing psychological disorders such as anxiety during their disease course, lipedema patients may be experiencing underlying difficulties in ER. For example, in an international survey with *N* = 1358 participants with lipedema, Clarke et al. (2022) showed that 28% reported emotional lability (rapid and often exaggerated changes in mood associated with powerful emotions). The high percentage of those with lipedema presenting with emotional lability suggests that there may be many lipedema patients who experience difficulties in managing their emotions [10].

Indeed, greater empirical attention could be directed to ER difficulties in patients with lipedema in improving QoL and understanding mental health concerns. To our knowledge, no studies investigate or evaluate the overall ER difficulties in patients with lipedema. Therefore, this study investigates the ER difficulties among patients with lipedema compared with those without lipedema. The current study assesses potential differences in ER abilities between patients with lipedema and healthy people without lipedema. We hypothesized that the lipedema patients’ group would demonstrate a deficit and severe difficulties in the ER compared to people without lipedema.

## 2. Materials and Methods

### 2.1. Study Design and Participants

The patients with lipedema were invited to participate from the San Giovanni Battista Hospital in Rome (Italy) and the Policlinico Tor Vergata hospital, Rome, Italy. Participants were screened for eligibility by a dermatologist consultant. Patients’ eligibility screening, recruitment, and inclusion in the study were obtained between April 2019 and January 2020. Inclusion criteria were: (1) A confirmed lipedema diagnosis by medical specialists at the San Giovanni Battista Hospital in Rome (Italy), (2) patients with lipedema at any stage or type. Exclusion criteria were: (1) History or presence of depression and anxiety diagnosis and use of psychiatric medications, (2) presence of a primary acute health-threatening disease (e.g., cancer, acute infection, or significant injury), (3) drug addiction, (4) pregnancy and breastfeeding. The control group (subjects without lipedema) was recruited from participants in the department who participated in the other studies active in the same period. We excluded any participant from the control group with severe health disease or acute illness or had a history of taking any psychiatric therapy.

All eligible participants provided written informed consent. The study was carried out according to the Declaration of Helsinki and was approved by the Local Ethics Committee.

### 2.2. Outcome Measures

Demographic information was obtained from participants, including age, sex, height, weight, and body mass index (BMI) were measured. In addition, the duration of lipedema and the clinical stages of lipedema that are based on morphological appearance were obtained by a specialist doctor. 

The ER abilities were assessed using the Italian version of the Difficulties in Emotion Regulation Scale (DERS) [22]. This 36-item self-report questionnaire produces a total ER score and a score in six subscales [23]. These subscales include: (1) Impulse control difficulties, (2) difficulties engaging in goal-directed behavior when emotionally aroused, (3) lack of emotional awareness, (4) lack of emotional clarity, (5) non-acceptance of emotions, and (6) limited access to emotion regulation strategies responses. The DERS was a reliable tool for assessing ER in different eating disorders [22]. The internal consistency of the DERS in our study was Cronbach’s alpha of 0.91. Scores are presented as a total score and a score for each of the 6 subscales. Higher scores suggest more significant problems with emotion regulation. This measure was used to capture transdiagnostic features of mental health related to emotional regulation.

The severity of anxiety was assessed by Hamilton-Anxiety (HAM-A) scale in patients with lipedema and healthy people [24]. HAM-A included 14 items distributed into two parts. The first part has seven items related to anxious mood symptoms, while the second part also has seven items related to physical symptoms of anxiety. The total score is obtained by a sum of the values (degrees) assigned to all 14 items on the scale. The anxiety levels according to the HAM-A are: None = 0; Mild = 1; Moderate = 2; Severe = 3; Very Severe = 4. The sum of the scores obtained on each item results in a total score ranging from 0 to 56. 

### 2.3. Statistical Analysis

As expected, the Shapiro–Wilk tests showed non-normality for three variables (BMI, anxiety, and the goal-directed behavior subscale of the DERS). Therefore, analyses considered robust to violations of normality were utilized. Statistical analyses were performed using the SPSS version 28 software package (IBM), with *p* < 0.05 considered significant. Descriptive statistics (means, standard deviations) were used to describe participant characteristics. Pearson’s product-moment correlations were used to determine the relationships between age, BMI, anxiety, and emotional regulation subscales. Analysis of variance (ANOVA) was used to determine group differences between those with lipedema compared to the control group. As preliminary analyses showed that BMI significantly differed between the two groups and significantly correlated with anxiety and emotional regulation subscales, a follow-up analysis of covariance (ANCOVA) was used to determine group differences (lipedema vs. control) in anxiety and emotional regulation scores after controlling for BMI.

## 3. Results 

### 3.1. Participant Characteristics and Clinical Variables

Out of 40 patients with lipedema screened, 14 were excluded (three did not provide consent, four were utilizing psychiatric medication use, three had a history of drug addiction, and four were breastfeeding mothers). A total of 26 were included in this study. Participant characteristics are presented in Table 1. On average, participants were 40.65 years old (*SD* = 13.40) and had a BMI of 27.39 (*SD* = 8.32). The lipedema group showed a significantly higher BMI than the control group (Table 1). The average (±*SD*) disease duration for lipedema patients was 8.65 ± 6.34 years. All patients had lipedema of the lower limb, but upper limb involvement was present in 12 patients. Thirteen patients had stage I lipedema, seven patients had stage II lipedema, and six patients had stage III lipedema. Eight lipedema patients reported that they received three procedures of liposuctions for their lower limbs. None had been diagnosed with psychiatric disturbances or cognitive impairments.

### 3.2. Analyses

As shown in Table 2, correlation analyses showed significant positive relationships between BMI, anxiety, and emotional regulation. Increasing BMI was related to more significant anxiety and difficulty with emotional regulation total score and subscales (excepting non-acceptance of emotions) (*p* < 0.01). Strong positive relationships were identified by Pearson’s correlations between anxiety and emotional regulation total scores on all subscales, with more significant anxiety related to greater emotional regulation difficulties (*p* < 0.001).

Differences between lipedema and control groups on anxiety and total and subscale emotional regulation scores are shown in Table 3. The means showed that those with lipedema had the most significant difficulty with emotional clarity, non-acceptance of emotions, and emotional regulation strategies. The ANOVA results show significantly higher levels of anxiety and difficulty in emotional regulation in the lipedema group compared to the control. When adjusted for BMI, ANCOVA results showed that group effects remained significant, with Cohen’s *d* demonstrating large effects across anxiety and difficulties in emotional regulation total scores and subscale scores (impulse control, goal-directed behavior, emotional awareness, emotional clarity, non-acceptance of emotions, and emotional regulation strategies), with the most significant effect shown for non-acceptance of emotions. 

## 4. Discussion

The present study is the first to investigate different dimensions of emotional regulation difficulties among patients with lipedema. The results of the present study demonstrated that lipedema patients have more difficulty regulating their emotions compared to those without lipedema. Specifically, our findings showed that the DERS scores and HAM-A are worse in the lipedema group, indicating that lipedema patients presented more difficulties in emotional regulation and more significant anxiety than people without lipedema. Interestingly, these results support our hypothesis that lipedema patients had significant deterioration in their ER. Further, while patients with lipedema are recognized or categorized as overweight or obese people [3,25], obese people have more ER difficulties compared with non-obese [26,27], differences in ER between lipedema patients compared to non-lipedema patients remained after accounting for BMI. Therefore, understanding the impact of ER difficulties and emotional distress for women with lipedema is essential for improving mental health outcomes, such as anxiety.

The current findings showed that emotional processing was impaired in lipedema patients compared to healthy controls. In particular, lipedema patients reported significantly more incredible difficulty in emotional awareness, particularly emotional clarity and acceptance of emotions. These results indicated that patients with lipedema may have significant difficulties in describing and understanding their emotions. While this is in line with research demonstrating that obese women exhibited deficits in emotional awareness [28], the current study showed these differences in emotional clarity and acceptance of emotions remain beyond the effects of BMI for lipedema patients compared to healthy control. These emotional processing abilities are vital to emotional regulation to cope with emotions. That is, one must be able to identify and accept the emotion they are experiencing to deal with it effectively.

A lack of clarity and acceptance of emotions reduces the likelihood of being able to choose appropriate behaviors and strategies to control these unfamiliar emotions. This may explain why lipedema patients reported more difficulty controlling impulsive behavior and their ability to engage in goal-directed behavior than healthy controls, as unconscious emotions undermine the control of desired behaviors. Whilst, again, previous studies described that both impulsiveness and negative urgency are linked directly with obesity or overweight and weight gain [29,30], results demonstrate effects of lipedema beyond BMI. ER plays an integral role in goal-directed behaviors and is critical to drawing human attention to significant life events [31,32]. It is essential to mention that with more significant difficulties in emotional clarity and impulse control and engaging in goal-directed behaviors, it is not surprising that patients with lipedema also report not having access to effective strategies to regulate their emotions at times of emotional distress. These results may explain why patients with lipedema do not use ER strategies, such as cognitive reappraisal, and many experience emotional lability. This finding is consistent with previous research that demonstrated that adolescents with anxiety and depression disorders have engaged in less reappraisal, problem-solving, acceptance, and avoidance than healthy individuals [33]. Thus, the results of the current study demonstrated that patients with lipedema have less access to ER strategies that they perceive effective.

In fact, physical and mental consequences may progress in patients with lipedema due to misdiagnosis and lack of appropriate treatment [3,4,8,10,34]. Given that a low ability to engage in goal-directed behavior is also known to be strongly associated with depressive and anxiety symptoms [35,36,37] and that lipedema patients have an increased risk of depression and anxiety [3], it is unsurprising then, that more significant difficulties in emotional regulation were associated with greater severity of anxiety symptoms in the current study. One explanation for why lipedema patients experienced more significant distress and difficulty beyond the influence of BMI may be the additional burden inherent in dealing with lipedema-related symptoms. For example, pain is a common disabling symptom [3,4] reported by patients with lipedema that significantly impacts the quality of life [3]. Further, pain has also been shown to play an important role in aggravating ER difficulties [38]. Despite the significant relation between ER and psychological disorders, it has been suggested that emotional dysregulation is a high-risk factor for the development of psychopathology [16,39]. The current study provides preliminary evidence to the theoretical speculation that the ER difficulties associated with lipedema may be a silent factor that enhances the risk of developing mental problems and may be strongly related to functional impairment among lipedema patients. In particular, this study showed that all dimensions of difficulties in emotional regulation were related to the severity of anxiety symptoms. However, this was strongest for emotional regulation strategies. Educational interventions teaching emotional regulation strategies may improve mental health outcomes for those with lipedema.

Psychotherapeutic interventions that aim to improve difficulties in emotional regulation have been shown to significantly reduce symptoms of anxiety and depression [40,41]. Those with lipedema may particularly benefit from those therapies and interventions that target emotional regulation and teach effective emotional regulation strategies. For example, mindfulness interventions, which teach skills in emotional clarity, awareness, and acceptance of emotions through training in attention and acceptance, have significantly improved anxiety and psychological well-being [42,43]. Further, therapies, such as Acceptance and Commitment Training (ACT) and Compassion-Focused Therapy (CFT), additionally target the acceptance of emotions, commitment to taking appropriate action and teach strategies to compassionately address painful and complex emotions to ease the difficulty [44,45].

This study has several limitations due to the cross-sectional design, the bias of ER evaluation (obtained by self-reported questionnaires), and the non-random sampling because of the statistical population. Moreover, other factors potentially affecting the outcomes were not included in the analysis, such as depression, mindful attention evaluation, pain severity assessment, and more sensitive indexes of cognition. Therefore, future studies should assess ER in patients with lipedema by using measures that do not require high levels of introspection. In addition, our control group is healthy, thus, recruiting a population with other chronic conditions with comparable clinical characteristics (e.g., fibromyalgia) is recommended in future studies. 

Despite these limitations, this study contributes to the literature being the first to our knowledge to assess ER difficulties in patients with lipedema. Understanding the impact of ER difficulties on those with lipedema provides a deeper psychological awareness of psychological changes that may enable earlier awareness of mental health concerns. Further, treatments, such as Mindfulness, Acceptance, and Commitment Therapy and Compassion-Focused Therapy, may offer potential strategies to assist in emotional distress and regulate fluctuating physical and emotional body and mental states. Lipedema is often misdiagnosed, so future studies measuring ER in patients with lipedema may provide needed validation and awareness of emotional difficulties that emerge in the progression and stage changes of lipedema. 

## 5. Conclusions

Overall, the present study’s findings indicate that patients with lipedema have difficulty controlling and regulating their emotions. These ER difficulties may increase the incidence of mental health-related problems, such as anxiety, in patients with lipedema. Our results are encouraging and have important implications for clinical and health psychologists. We recommended health care providers pay more attention to ER difficulties and general psychological disorders in lipedema patients.

## Figures and Tables

**Table 1 ijerph-19-13679-t001:** Participant Characteristics.

	LIP *N* = 26	CG *N* = 26	*F* (1, 50)
Age (years) (*M*, *SD*)	40.73 (11.01)	40.58 (15.65)	0.00
BMI (kg/m^2^) (*M*, *SD*)	30.50 (9.49)	24.27 (5.57)	8.34 **
Duration of illness (years) (*M*, *SD*)	2.81 (1.65)		
Lipedema stage (*n*, %)			
Stage I	13 (50.0%)		
Stage II	7 (26.9%)		
Stage III	6 (23.1%)		

Note. Abbreviations: LIP: Lipedema group, CG: Control group (healthy people without lipedema), BMI: Body mass index. ** *= p* < 0.01.

**Table 2 ijerph-19-13679-t002:** Pearson’s Product Moment Correlations Between Demographics, Anxiety, and Emotional Regulation.

	1	2	3	4	5	6	7	8	9
1. Age									
2. BMI	0.13								
3. Anxiety	0.06	0.39 **							
4. Emotional regulation total score	0.00	0.38 **	0.84 ***						
5. Impulse Control	−0.08	0.32 *	0.73 ***	0.89 ***					
6. Goal-Directed Behaviour	0.09	0.30 *	0.73 ***	0.88 ***	0.74 ***				
7. Emotional Awareness	−0.05	0.37 **	0.64 ***	0.86 ***	0.75 ***	0.75 ***			
8. Emotional Clarity	0.01	0.30 *	0.80 ***	0.91 ***	0.76 ***	0.78 ***	0.68 ***		
9. Non-Acceptance of Emotions	−0.01	0.18	0.86 ***	0.91 ***	0.76 ***	0.76 ***	0.70 ***	0.85 ***	
10. Emotional Regulation Strategies	0.02	0.58 ***	0.77 ***	0.90 ***	0.80 ***	0.71 ***	0.75 ***	0.78 ***	0.76 ***

* *p* ≤ 0.05, ** *p* ≤ 0.01, *** *p* ≤ 0.001.

**Table 3 ijerph-19-13679-t003:** Analysis of Covariance: Group Differences in Anxiety and Difficulties in Emotional Regulation Controlling for BMI.

				Unadjusted Main Effects	Adjusted Main Effects	
	LIP	CG	Total	*F* (1, 50)	*F* (1, 49)	Cohen’s *d*
Anxiety (HAM-A) (*M*, *SD*)	27.62 (8.98)	4.96 (2.51)	16.29 (13.17)	153.64 ***	123.10 ***	3.17
Emotional regulation (DERS)						
Total score (*M*, *SD*)	135.69 (13.12)	53.00 (9.03)	94.35 (43.21)	700.45 ***	582.95 ***	6.88
Impulse control	20.23 (4.81)	7.77 (2.67)	14.00 (7.38)	133.30 ***	112.14 ***	3.01
Goal-directed behavior	21.00 (4.13)	8.62 (2.59)	14.81 (7.12)	167.80 ***	147.07 ***	3.46
Emotional awareness	20.69 (5.42)	9.15 (3.02)	14.92 (7.27)	89.87 ***	71.31 ***	2.41
Emotional clarity	24.77 (5.28)	9.27 (3.12)	17.02 (8.93)	166.18 ***	144.66 ***	3.43
Non-acceptance of emotions	24.69 (5.09)	8.19 (3.36)	16.44 (9.36)	190.37 ***	212.98 ***	4.17
Emotional regulation strategies	24.31 (4.83)	10.00 (3.32)	17.15 (8.31)	154.81 ***	142.31 ***	3.41

Note. DERS: Difficulties in Emotion Regulation Scale, HAM-A: Hamilton Anxiety scale. **** p* ≤ 0.001.

## Data Availability

The data that support the findings of this study are available from the corresponding author upon reasonable request.

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
