# Peer review of "The Difficulties in Emotional Regulation among a Cohort of Females with Lipedema"

_ijerph, 2022, doi:10.3390/ijerph192013679_

Round 1
Reviewer 1 Report
The manuscript presents the analysis of emotional regulation and anxiety in patients with lipedema as compared to healthy control subjects. Several limitations of the study are reported in the discussion. Although further studies are definitely needed to better investigate the influence of emotional regulation in the onset and progression of the disease, this study provides useful and valuable results that needs to be taken into consideration in the clinical practice. Minor revisions are needed and listed below:
1. The affiliation 7 is not associated to any of the authors listed. Please check it.
2. There’s no background in the abstract. A brief introduction (one or two sentences) should be added and precede the aim of the study.
3. Line 32, correct lipoedema with lipedema to be consistent trough the text.
4. Line 34-35, this is not the conclusion of the study, it’s more an interpretation that should be moved to the discussion.
5. Line 41, please include (Herbst, 2012) in the numbered reference list as [1].
6. Line 62, please correct “f” with “of”.
7. Line 78, please add the references (Cengiz & Gürel, 2020; Willem et al., 2020) in the numbered reference list.
8. Please cite this recently published paper (PMID: 34252343 DOI: 10.1080/07399332.2021.1932894). This paper is an interesting addition to the discussion provided here.
Author Response
The manuscript presents the analysis of emotional regulation and anxiety in patients with lipedema as compared to healthy control subjects. Several limitations of the study are reported in the discussion. Although further studies are definitely needed to better investigate the influence of emotional regulation in the onset and progression of the disease, this study provides useful and valuable results that needs to be taken into consideration in the clinical practice. Minor revisions are needed and listed below:
We gratefully thank this reviewer for their comments and we really appreciate the time taken to provide comments and criticisms that have been useful for improving our manuscript. Following is a point-by-point reply to all the criticisms raised. The new text is highlighted in yellow in the manuscript file.
- The affiliation 7 is not associated to any of the authors listed. Please check it.
RE: We apologize for this mistake, this affiliation is for Dr. Marco Marchetti, and we added it.
- There’s no background in the abstract. A brief introduction (one or two sentences) should be added and precede the aim of the study.
RE: We thank the reviewer for this suggestion. Accordingly, we added a brief introduction to the abstract and adjusted further text to accommodate the word count.
- Line 32, correct lipoedema with lipedema to be consistent trough the text.
RE: Thank you for this attention to detail – whilst both spellings are used, we have ensured that “lipedema” has been used consistently throughout.
- Line 34-35, this is not the conclusion of the study, it’s more an interpretation that should be moved to the discussion.
RE: We agree with this suggestion. The sentence is moved to the discussion section.
- Line 41, please include (Herbst, 2012) in the numbered reference list as [1].
RE: thank you for the detail – this reference has been included.
- Line 62, please correct “f” with “of”.
RE: Thank you – this typographical error has been corrected.
- Line 78, please add the references (Cengiz & Gürel, 2020; Willem et al., 2020) in the numbered reference list.
RE: We thank the reviewer for this suggestion, we corrected both references.
- Please cite this recently published paper (PMID: 34252343 DOI: 10.1080/07399332.2021.1932894). This paper is an interesting addition to the discussion provided here.
RE: We appreciated this suggestion. The paper has been added to the introduction section.

Reviewer 2 Report
Description of control group is weak/missing. Statisitical analysis is not appropriate. At least scatterplots (with group identiefiers) have to be given.
The inclusion critera should be reconsidered (14 out or 40 were excluded), maybe change the analysis (nn parametric) to 3 groups.
Ancova is not appropriate in the current case (see scatter plots, which I guess are two group spurious corrlations).
Selection of ER has to be justified.
Author Response
Description of control group is weak/missing. Statisitical analysis is not appropriate. At least scatterplots (with group identiefiers) have to be given.
Thank you for this comment, further details have been added about the control group, on pg.3 lines 108-111. Detailed response to the scatterplot comment we have provided below.
“The control group (subjects without lipedema) were recruited from participants in the department who participate in the other studies active in the same period. We excluded any participant from the control group with severe health disease or acute illness or have a history of taking any psychiatric therapy”.
The inclusion criteria should be reconsidered (14 out or 40 were excluded), maybe change the analysis (nn parametric) to 3 groups.
Thank you to the reviewer for this comment. It is unclear how and in what way the reviewer is intending to suggest three groups for analysis. This point of the research is to assess differences in emotional regulation between women with lipedema and a matched group of women without lipedema without the complication of other serious health or mental health concerns.
Ancova is not appropriate in the current case (see scatter plots, which I guess are two group spurious corrlations).
The overall findings of this research show clear links between lipedema status (control vs. lipedema) and emotional regulation over and above BMI, which is the main research question. ANCOVA was considered appropriate for accounting for the variance and ratio of likely relationship/difference in emotional regulation between population groups (control vs lipedema). Also, whilst different scatterplots may be helpful, they are a pictorial representation of the correlations already reported in Table 2. As such, the authors propose that this level of relationship has already been described with the Pearson r values, and as such the overall statistical approach the reviewer has suggested is preference only as opposed to a statistical limitation.
Selection of ER has to be justified.
Thank you for this comment. Further clarity has been added to Page 3 Lines 129
“This measure was used to capture transdiagnostic features of mental health related to emotional regulation”
